# Myricetin as a Promising Molecule for the Treatment of Post-Ischemic Brain Neurodegeneration

**DOI:** 10.3390/nu13020342

**Published:** 2021-01-24

**Authors:** Ryszard Pluta, Sławomir Januszewski, Stanisław J. Czuczwar

**Affiliations:** 1Laboratory of Ischemic and Neurodegenerative Brain Research, Mossakowski Medical Research Institute, Polish Academy of Sciences, 02-106 Warsaw, Poland; sjanuszewski@imdik.pan.pl; 2Department of Pathophysiology, Medical University of Lublin, 20-090 Lublin, Poland; czuczwarsj@yahoo.com

**Keywords:** brain ischemia, myricetin, amyloid, tau protein, autophagy, metal ion, oxidative stress, neuroinflammation, acetylcholine, neurodegeneration, dementia, therapy

## Abstract

The available drug therapy for post-ischemic neurodegeneration of the brain is symptomatic. This review provides an evaluation of possible dietary therapy for post-ischemic neurodegeneration with myricetin. The purpose of this review was to provide a comprehensive overview of what scientists have done regarding the benefits of myricetin in post-ischemic neurodegeneration. The data in this article contribute to a better understanding of the potential benefits of myricetin in the treatment of post-ischemic brain neurodegeneration, and inform physicians, scientists and patients, as well as their caregivers, about treatment options. Due to the pleiotropic properties of myricetin, including anti-amyloid, anti-phosphorylation of tau protein, anti-inflammatory, anti-oxidant and autophagous, as well as increasing acetylcholine, myricetin is a promising candidate for treatment after ischemia brain neurodegeneration with full-blown dementia. In this way, it may gain interest as a potential substance for the prophylaxis of the development of post-ischemic brain neurodegeneration. It is a safe substance, commercially available, inexpensive and registered as a pro-health product in the US and Europe. Taken together, the evidence available in the review on the therapeutic potential of myricetin provides helpful insight into the potential clinical utility of myricetin in treating neurodegenerative disorders with full-blown dementia. Therefore, myricetin may be a promising complementary agent in the future against the development of post-ischemic brain neurodegeneration. Indeed, there is a scientific rationale for the use of myricetin in the prevention and treatment of brain neurodegeneration caused by ischemia.

## 1. Introduction

With the aging of the world population, ischemic stroke has become the second leading cause of death in people aged 60 and over, and the fifth leading cause of death in people aged 15 to 59 worldwide [1]. Worldwide, 70% cases of ischemic stroke and 87% of deaths related to ischemic stroke and disability adjusted life years occur in low- and middle-income countries [2]. The incidence of ischemic stroke cases has more than doubled in low- and middle-income countries in the last four decades [2]. Over these decades, the incidence of ischemic stroke cases has decreased by 42% in high-income countries [2]. Ischemic stroke occurs on average 15 years earlier in people living in low and middle-income countries and causes more deaths compared to those in high-income countries [3]. As many as 84% of ischemic stroke patients in low- and middle-income countries die within three years of stroke, compared with 16% in high-income countries [2]. Current epidemiological statistics indicate that approximately 17 million people suffer from ischemic brain injury annually, of whom 6 million die each year [4,5]. Worldwide, the number of post-ischemia cases has now reached approximately 33 million [4,5]. According to the latest forecasts, the number of cases will increase in the future to 77 million in 2030 [4,5]. In 2010, the annual cost of treating stroke patients in Europe was approximately € 64 billion [5]. If the trend in ischemic stroke continues, there will be approximately 12 million deaths by 2030, 70 million will be stroke survivors, and more than 200 million disability-adjusted life years will be recorded worldwide annually [5].

The mechanisms of post-ischemic neurodegeneration are complex and unclear, and the development of it is influenced by many factors. Excessive amyloid accumulation and increased tau protein hyperphosphorylation are currently the most studied factors in post-ischemic neurodegeneration with dementia [6,7,8]. It is also believed that metabolic imbalances in the brain, such as hyperactivity of the glutamate system, acetylcholine deficiency and metal ion dyshomeostasis, are closely related to the development of neurodegeneration as a consequence of ischemic brain injury [9,10,11,12]. Moreover, abnormal processes such as oxidative stress, neuroinflammation, and impaired autophagy have been found to cause severe brain damage and contribute to progressive and irreversible damage following reversible ischemia [13,14,15,16,17,18,19]. Dyshomeostasis of the intestinal flora is also mentioned as the driving force in the genesis and development of neurodegeneration after ischemia [20,21].

In the treatment of stroke, the first priority is to focus on physical recovery, but in the first year after stroke, 4 out of 10 patients have some degree of cognitive impairment without dementia [22]. This may be related to demographic and disease factors. About 6 months after ischemic stroke, women with a history of cerebrovascular disease and patients with lacunar infarction develop new cognitive impairment [23]. Identification of dementia immediately after stroke is difficult because of additional persistent deficits after ischemic stroke, both in terms of global cognition and individual domains, e.g., attention and processing speed, memory, language, and frontal executive functions may be impaired [24]. Finally, it is true that a history of ischemic stroke is a strong independent risk factor for the development of dementia [8,24,25,26,27,28]. Brain ischemia accelerates the onset of dementia by 10 years [29], and 10% of people develop dementia soon after the first incidence of ischemic stroke and 41.3% after the recurrent stroke [24,30].

A transient episode of cerebral ischemia in humans and animals causes acute massive neuronal loss in the CA1 region of the hippocampus and the cerebral cortex [6,7,31,32,33,34,35,36]. In these structures, necrotic and apoptotic neurons were mixed with neurons damaged after ischemia [6,31,32,35,36]. After a short-term post-ischemic survival, the number of damaged neurons decreased while the number of dead neurons increased [31,32]. After prolongation of post-ischemia survival, acute and chronic neuronal changes were observed in ischemia-resistant areas, in addition to acute neuronal death in ischemia-sensitive areas like hippocampal CA1 subfield [31,32,33,34]. The lesions were present in areas of the brain not affected by the primary ischemic injury, such as the CA2, CA3, and CA4 areas of the hippocampus [31,32,33,34,35,36]. Following ischemia, neuronal loss was observed with a decrease in acetylcholine levels in the hippocampus, suggesting that an additional cause of neuronal death was a deficiency in neuronal excitation [9,12,37,38]. In addition, ultrastructural studies revealed changes in hippocampal synapses after ischemia [38,39]. Other observations showed that an episode of brain ischemia with recirculation led to the induction of synaptic autophagy, which may be associated with the death of pyramidal neurons in the hippocampus after transient reversible cerebral ischemia [9,16,17,18,37,38,39,40,41,42].

Alterations in white matter and activation of glial cells in the brain tissue have been reported in both humans and animals following ischemia with recirculation [13,19,31,32,43,44,45]. In animal models of reversible cerebral ischemia, an ischemic episode causes severe changes in the corpus callosum and subcortical white matter [32,43,44,46,47,48]. These alterations are consistent with the activation of astrocytes and microglia in the brain tissue after ischemia [13,19,49]. Late brain white matter atrophy in experimental animals manifested as progressive spongiosis. Cerebral ischemic lesions showed signs of progressive neurodegeneration that developed slowly over a long period of time in post-ischemic survival [32]. A brain autopsy 1–2 years after experimental ischemia revealed cerebral hydrocephalus with the widening of the ventricles and the subarachnoid space [31,32,46,47]. This was accompanied by the complete atrophy of the hippocampus with a very narrow cortex [31,32,46,47,50]. The ultimate consequence of this neuropathology was the development of dementia in experimental and clinical studies after transient and reversible cerebral ischemia [45,47,51,52,53,54,55,56].

It has been found that neurodegenerative processes develop not only in the acute stage of ischemia, but also progress throughout the survival period after ischemia [32]. Brain neurodegeneration seen after ischemia shares features and mechanisms with neurodegeneration seen in Alzheimer’s disease [5,8,14,15,25,47,48,57,58,59,60,61]. This is confirmed by the increased permeability of the blood–brain barrier after ischemia to inflammatory cells and the leakage of amyloid and tau protein from the blood serum into the brain parenchyma, which in turn probably leads to irreversible and progressive damage to the whole brain [13,19,43,44,55,56,57,62,63,64,65,66,67,68,69,70,71,72,73,74,75]. Mental health deterioration and the onset of post-ischemic neurodegeneration-related cognitive impairment have raised concerns and triggered serious scientific debate. Progressive ischemic neurodegeneration of the brain has been found to be associated with the overproduction of folding proteins such as amyloid and tau protein [5,6,7,47,48,57,59,60,76,77,78,79,80]. Diffuse and senile amyloid plaques were found in the hippocampus and brain cortex after ischemia [57,81,82,83,84,85,86]. Data indicate that cerebral ischemia is involved in the development of paired helical filaments [87], neurofibrillary tangle-like [88,89,90], and neurofibrillary tangles [91,92] after this event. In this regard, special attention has been paid to the role of amyloid and tau protein as additional new contributing factors in the development of post-ischemic dementia [6,7,8,25,55,56,59]. Recently, changes in proteins and genes associated with Alzheimer’s disease following ischemic brain injury and their possible role in ischemic brain neurodegeneration were presented. New advances in understanding the development of post-ischemic neurodegeneration have revealed a dysregulation of the genes for amyloid protein precursor, α-secretase, β-secretase, presenilin 1 and 2, tau protein, autophagy, mitophagy, and apoptosis [7,16,17,18,93,94,95,96]. Ultimately, Alzheimer’s disease-related proteins and their genes have been documented to play an important role in the progression of post-ischemic brain neurodegeneration with subsequent development of dementia [8].

Due to the loss of neurons in the brain and the associated loss of neuronal network function, brain ischemia is also the leading cause of permanent disability in adults worldwide, reducing patient quality of life and increasing the global medical burden [2]. Early neuropsychiatric symptoms and dementia following ischemic stroke increase the risk of mortality and recurrence of ischemic stroke [97,98,99]. Despite its enormous impact on the socio-economic development of countries, this growing problem has so far received very little attention. Current management and treatment of most of the sequelae of ischemic stroke are unsatisfactory, with the exception of some antidepressants, which have therapeutic benefit [1].

In the absence of a translation of experimental neuroprotective substances for clinical use [100], the patients are very interested in improving motor and cognitive functions after ischemia, and not in protecting ischemic neuronal death in the brain. Therefore, we are now forced to improve the survival of persistent neurons and the associated cognitive functions after ischemia [27,51,52,53,54]. New treatments should improve activity following ischemic brain injury via effectively extending the therapeutic window. It should be emphasized that, despite the fact that ischemic stroke is one of the leading causes of death and disability worldwide, there has recently been a lack of effective post-stroke causal therapies that heal the structural and functional injuries, ultimately leading to post-ischemic neurodegeneration of the brain with subsequent developmental dementia. Therefore, in this review we will focus on the protective effect of pleiotropic myricetin on persistent neurons and neuropathological phenomena that develop after ischemic stroke and experimental cerebral ischemia.

## 2. Myricetin

3,3′,4′,5,5′,7-hexahydroxyflavone (myricetin) (Figure 1) is a flavonoid that was first identified in *Myricaceae* plants about 2 centuries ago [101,102]. Myricetin is light yellow in color. Commonly consumed fruits and vegetables are rich in myricetin. Strawberries, spinach, apples, aloe, carrots and mulberries are rich in myricetin, and the myricetin content in red wine is twice that of resveratrol [103,104,105,106]. In addition, myricetin has been approved as a health product in US by Food and Drug Administration and in Europe and has been successfully introduced to the general market [107]. Moreover, myricetin is an essential component of healthy food and drink and has an excellent safety profile combined with the possibility of human consumption [101]. Being one of the more studied polyphenols, myricetin has a number of biological properties [102]. Myricetin has been shown to have anti-oxidant, anti-inflammatory and anti-tumor effects [102,108,109]. Myricetin has been found to have therapeutic properties in Alzheimer’s disease [102]. It is believed that aloe and mulberry rich in myricetin have additional anti-dementia effects [102,110,111].

## 3. Possible Use of Myricetin in Post-Ischemic Neurodegeneration

Transient ischemic-reperfusion brain injury in humans and animals is predominantly age-related and is characterized by various cognitive impairments, but primarily memory deficits, with gradual cognitive and intellectual decline, ultimately leading to the full-blown dementia [27,51,52,53,54]. In elderly humans, it is the leading cause of death. The accumulation of diffuse and senile amyloid plaques in the brain and neurofibrillary tangles in neuronal cells are new phenomenona in patients with cerebral ischemia and in animals post-ischemia [57,81,82,83,84,85,86,87,88,89,90,91,92]. It is believed that the deposition of misfolded proteins is an additional cause of neuronal death, loss of synapses, oxidative damage, and the development of neuroinflammation in the brain following ischemia [13,19,35,36,47,48]. Therefore, the greatest medicinal potential for neurodegeneration after stroke and animal ischemia should have molecules with pleiotropic activity, especially those with anti-amyloid, anti-tau protein and anti-dementia properties, as well as those that reduce oxidative stress and neuroinflammation [102]. Recently, myricetin is considered to be one of the most interesting and promising natural pleiotropic substance for use in the treatment of stroke due to its pleiotropic effects [102]. Myricetin is a substance with strong anti-oxidant and anti-inflammatory properties [102]. Its anti-amyloid and anti-tau protein properties make it the most promising compound for the treatment of various neurodegenerative disorders involving the deposition of folding proteins [102]. Neuroinflammation, oxidative damage, and misfolded protein deposition have been shown to synergistically contribute to post-ischemic brain injury [13,19,57,81,82,83,84,85,86,87,88,89,90,91,92]. Therefore, targeting these phenomena may be an essential strategy in the treatment of post-ischemic brain neurodegeneration. In this context, the use of myricetin in the treatment of ischemic neurodegeneration has certain advantages: namely, it (1) decreases the production of amyloid, (2) prevents the development of amyloid oligomers and fibrils, (3) protects the development of neurofibrillary tangles, (4) decreases neuroinflammation, (5) acts as a powerful anti-oxidant, (6) prevents metals from binding to amyloid and tau protein, (7) increases acetylcholine levels (Figure 1), and (8) can be taken in relatively high quantities without side effects [102].

Brain injury following ischemia is a neurodegenerative disease that can cause patients to gradually lose their ability to live independently and change their personality and behavior. Survivors of cerebral ischemia suffer from cognitive and linguistic deterioration, including speech and memory. Ischemic stroke not only threatens the life and health of patients, but also causes serious social problems, especially in countries with an aging population [2]. Recently, neuropsychiatric problems and post-stroke dementia have been of concern to clinicians and researchers. Mental problems and dementia are common post-stroke complications and are associated with poorer outcomes, including poor quality of life, increased burden of care, and poor functional status [112,113]. Studies have shown that myricetin ameliorates cognitive dysfunction by promoting amyloid clearance and inhibiting neuroinflammation in animal models of Alzheimer’s disease [110,111,114]. In this section of the review, we present a possible use of myricetin in the treatment of post-ischemic brain neurodegeneration that has neuropathological changes similar to Alzheimer’s disease.

## 4. Myricetin versus Amyloid

Following cerebral ischemia, β- and γ-secretase are involved in the production of amyloid from the amyloid protein precursor [93,94,95,96]. The overproduction and increase deposition of amyloid in the brain following ischemia [31,32,57] are associated with the onset and progression of neuropathological changes characteristic of Alzheimer’s disease. It has been shown that myricetin inhibits the activity of β-secretase, thus decreasing the production of amyloid [115]. Moreover, myricetin has been shown to increase the level of α-secretase, which results in an increased level of harmless fragments of the amyloid protein precursor [115]. This results in an overall decrease in the levels of the amyloid protein precursor that is used for amyloid production, thereby indirectly decreasing amyloid generation, too. The amyloid monomer exhibits neurotrophic activity, but amyloid oligomers and fibrils show strong neurotoxicity through increased neuroinflammation, destruction of cell membranes and oxidative stress [116,117,118]. Amyloid oligomers and fibrils are produced by excess amyloid through the β-sheet [116], and myricetin inhibits β-sheet formation [102,115,119]. Moreover, myricetin may bind to amyloid fibrils, thus inhibiting amyloid fibril elongation and maturation and the formation of diffuse and senile amyloid plaques (Figure 1) [102,120,121]. Thus, myricetin hinders the formation of amyloid oligomers, which decreases the neurotoxicity of amyloid and decreases the progression of ischemic pathological damage.

## 5. Myricetin versus Autophagy

Autophagy takes place within lysosomes to eliminate worn-out organelles or damaged proteins [41]. Autophagy and mitophagy proteins and gene alterations are observed in post-ischemic brain neurodegeneration [8,16,17,18]. In the case of neurons, it is difficult to get rid of, for example, toxic proteins (amyloid, dysfunctional tau protein) by cell division, so autophagy is a particularly important process for these cells. At the beginning of the autophagy, the endoplasmic reticulum in the neuron forms autophagic vesicles. These autophagic vesicles surround harmful proteins such as amyloid and incorrectly phosphorylated tau protein to form autophagosomes. Then the autophagosomes are transported along the microtubules by kinesin to the lysosomes, where the degradation of toxic proteins by lysosomes takes place [42,122]. In addition, current studies indicate that mammalian targets of rapamycin are a key signaling factors controlling cell proliferation and apoptosis and a major cytokine increasing cellular autophagy. Myricetin influences the increase of autophagy (Figure 1), which results in the elimination of toxic amyloid and the dysfunctional tau protein produced by neurons [102,123]. Myricetin induces protective autophagy by inhibiting the phosphorylation of mammalian targets of rapamycin, and this effect is dose dependent [102,123].

## 6. Myricetin versus Metal Ions

Disruption of the balance of metal ions in an ischemic brain can result in cytotoxicity, oxidative stress, and increased amyloid deposition—changes closely related to post-ischemic brain neurodegeneration [9,10,11,14]. Initially, studies on metals associated with ischemia focused on calcium ions [9], but in recent years attention has been focused on other metal ions such as zinc (Zn), copper (Cu), and iron (Fe), which are associated with the development of post-ischemic neurodegeneration [9]. Observations have shown that there are several binding sites within amyloid that bind metal ions, and a significant increase in amyloid toxicity is seen when this complexation occurs. Consider Zn^2+^ which has binding sites in amyloid, so even at micromolar concentrations it increases amyloid aggregation [124]. Myricetin has the ability to chelate metal ions, which may inhibit the effects of ischemia by controlling the concentration of metal ions in the brain [125]. It has been shown that myricetin can inhibit amyloid aggregation through chelation of Cu^2+^ or Zn^2+^ [126]. Myricetin can control the level of metal ions in an ischemic brain by forming complexes with metal ions, which decreases the likelihood of amyloid binding to metal ions. Moreover, myricetin may not only prevent binding of amyloid to metal ions, but also break down the resulting Zn^2+^/Cu^2+^ amyloid complexes [126]. In addition to affecting amyloid aggregation, Zn^2+^ may also have an effect on amyloid generation (Figure 1). Zinc can increase the level of β- and γ-secretase by inhibiting the activity of α-secretase, thus affecting the level of generated amyloid in the brain [127]. This indicates that the complexation of myricetin and Zn^2+^ may also decrease the level of amyloid. Moreover, like other divalent ions, Fe^2+^ can bind amyloid and trigger the development of amyloid oligomers and fibrils [128]. Upon complexation of Fe^2+^ by myricetin, the level of free Fe^2+^ decreases, which decreases the Fenton reaction converting H_2_O_2_ into highly toxic hydroxyl radicals and reactive oxygen species (ROS) [129,130]. This can lead to less damage from oxidative stress in an ischemic brain (Figure 1). Additionally, Fe^2+^ may lead to microglia activation, resulting in brain damage due to the development of neuroinflammation [131]. The effect of myricetin on Fe^2+^ may decreases the risk of neuroinflammation (Figure 1). Myricetin may also decrease iron levels by inhibiting the expression of the transferrin receptor 1 [132].

## 7. Myricetin versus Oxidative Stress

Oxidative stress is one of the direct mechanisms involved in the development of ischemic neurodegeneration of the brain [14]. Physiologically, the brain has high oxygen consumption but low antioxidant capacity, which makes it particularly susceptible to oxidative stress [133]. Free radicals and ROS are the two main mechanisms used by oxidative stress to injure neurons. By combining with radicals myricetin creates stable semiquinone radicals, breaking the radical chain reaction. In vitro experiments have shown that dose-dependent myricetin can effectively inhibit the formation of ROS and protect neurons from injury caused by oxidative stress (Figure 1) [134]. The effect of this is direct and indirect restoration of the physiological levels and activity of antioxidants such as catalase activity, superoxide dismutase (SOD), and glutathione peroxidase in cells [135,136]. Furthermore, in the H_2_O_2_-induced cell damage model, myricetin prevents oxidative stress-induced damage to DNA and lipids by regulating mitogen-activated protein kinase and phosphatidylinositol 3-kinase/protein kinase B signaling pathways. This leads to an increase in the level of anti-apoptotic molecules such as Bcl-2 and a decrease of pro-apoptotic mediators such as Bax, caspase-9, and caspase-3, resulting in inhibition of apoptosis induced by oxidative stress in cells (Figure 1) [135,137]. In addition to its direct effect on brain injury, oxidative stress is interrelated with factors associated with ischemic brain alteration, such as amyloid, which is capable of inducing oxidative stress, and oxidative stress induces amyloid production [138,139]. Myricetin inhibits the free radical chain reaction by inhibiting the amyloid source, thus decreasing the brain injury caused by oxidative stress [115]. By activating the c-Jun N-terminal kinase/stress-activated protein kinase (JNK/SAPK) pathway, oxidative stress increases the level of β-secretase. β-secretase metabolizes amyloid protein precursor to amyloid, leading to an increase in amyloid levels in ischemic brain [57,93,96], and an increase in amyloid levels further activates the JNK/SAPK pathway leading to a vicious cycle. Thus, myricetin may protect ischemic neurons from death due to its own antioxidant properties.

## 8. Myricetin versus Neuroinflammation

Investigations indicate that neuroinflammation is also one of the major processes of ischemic neurodegeneration, although neuroinflammation is to be the result of ischemic brain damage [13,19]. Myricetin directly decreases neuroinflammatory intensification (Figure 1), by inhibiting microglia activation and nucleotide-binding oligomerization domain-like receptor protein 3 [102,140]. Ultimately, these anti-inflammatory effects of myricetin help to ameliorate the severity of post-ischemic pathology in the brain (Figure 1). Myricetin can decrease the level of inflammatory factors such as interleukin (IL), tumor necrosis factor-alpha (TNF-α), and nuclear factor kappa B (NF-κB) [136,140]. Considering interleukin-1 (IL-1), it can not only damage neurons, but also elevate the level of the amyloid protein precursor, which increases the generation and accumulation of amyloid in the brain [141]. On top of it, IL-1 also increases TNF-α, which exacerbates the neuroinflammatory response and increase the cell and brain damage [141]. The anti-inflammatory activity of myricetin is mainly by decreasing the levels of IL, TNF-α, inducible nitric oxide synthase and cyclooxygenase-2 in the brain by disrupting the NF-κB signaling pathway and mitigating the damage caused by these factors [140].

Activated microglia play an important role in the development of post-ischemic neuroinflammation in the brain tissue [13,19]. Activated microglia occurs in two forms, M1 and M2: M1 promotes the development of neuroinflammation of the brain, while M2 inhibits the development of neroinflammation [142]. Finally, in the pathological process, the nucleotide-binding oligomerization domain-like receptor protein 3 in the brain can be activated by amyloid which can exacerbate neuropathological processes [143]. Studies show that amyloid levels can be effectively lowered by inhibiting nucleotide-binding oligomerization domain-like receptor protein 3, and that spatial memory impairment can be ameliorated in this situation (Figure 1) [143,144]. Currently, nucleotide-binding oligomerization domain-like receptor protein 3 in the brain is considered an effective target in the therapy of neurodegenerative disorders [145]. Moreover, myricetin may inhibit the activation of nucleotide-binding oligomerization domain-like receptor protein 3 by inhibiting apoptosis-associated speck-like proteins oxygen-dependent ubiquitination and promoting oxygen-independent ubiquitination of nucleotide-binding oligomerization domain-like receptor protein 3 [146]. The inhibitory effect of myricetin on nucleotide-binding oligomerization domain-like receptor protein 3 may decrease post-ischemic neuroinflammation and, to some extent, decrease amyloid levels in the brain (Figure 1).

## 9. Myricetin versus Acetylcholine

Acetylcholine is a neurotransmitter that plays a key role in the transmission of neural signals and memory formation, and the absence of acetylcholine in the brain, especially in the hippocampus is a known result of dementia in post-ischemic neurodegeneration [12]. Myricetin has been shown to be effective in inhibiting acetylcholinesterase, which breaks down acetylcholine in the brain [132,147]. Myricetin was effective in reducing the learning and memory impairment in aging brain through its ability to inhibit acetylcholinesterase (Figure 1) [148]. In addition to acetylcholinesterase, some inflammatory factors such as IL-1 also affect acetylcholine levels. IL-1 may increase the level of acetylcholinesterase and accelerate the breakdown of acetylcholine, causing insufficient content of acetylcholine in the brain and affecting the ability to remember [141]. The anti-inflammatory ability of myricetin may also indirectly prevent acetylcholine loss.

## 10. Myricetin versus Tau Protein

Disruption of the balance of metal ions in an ischemic brain can result in cytotoxicity, oxidative stress, increased amyloid deposition and tau protein hyperphosphorylation, changes closely related to post-ischemic brain neurodegeneration [9,10,11,14]. Zinc which has binding sites in tau protein, even at micromolar concentrations increases abnormal tau protein conformation (Figure 1) [124]. It has been shown that myricetin can inhibit tau protein hyperphosphorylation, through chelation of Zn^2+^ [126]. Oxidative stress also promotes tau protein hyperphosphorylation by inhibiting protein phosphatase 2A, and the dysfunctional tau protein increases oxidative stress and finally destroying synapses and mitochondria in a vicious circle [138,149,150,151]. IL-1 can also trigger tau protein hyperphosphorylation in the brain with further formation of neurofibrillary tangles, and neurofibrillary tangles are one of the major pathological elements of post-ischemic brain neurodegeneration in addition to amyloid (Figure 1) [91,92,141]. The anti-oxidative and anti-inflammatory ability of myricetin may also indirectly prevent tau protein hyperphosphorylation.

## 11. Myricetin versus Experimental Brain Ischemia

The aim of preliminary preclinical studies was to investigate whether myricetin may prevent damage from local brain ischemia and to identify potential mechanisms involved [136,152]. The rate of neurological deficit and the area of infarction caused by focal brain ischemia decreased in a dose-dependent manner after treatment with myricetin [136,152]. In addition, myricetin decreased the levels of interleukin-6, interleukin-1β, malondialdehyde, and TNF-α and increased the glutathione/glutathione disulfide ratio and SOD action [136]. In response to myricetin, a significant decrease in neuronal death was observed [136,152]. Moreover, myricetin significantly increased the level of phosphorylated protein kinase B (AKT) and decreased the phosphorylation of p38 mitogen-activated protein kinase (p38 MAPK) and the level of nuclear factor kappa B/p65 (NF-κB/p65) [136]. Taken together, the observations from these studies suggest that myricetin has a neuroprotective effect by decreasing ischemic brain damage via enhancement of the activity of antioxidant enzymes and improvement of mitochondrial function and that the protective effect of myricetin may be related to inactivation of the p38 MAPK and NF-κB/p65 pathways and activation of the AKT and nuclear factor erythroid 2-related factor 2 pathways [136,152].

## 12. Conclusions

Damage and loss of neurons, with the accumulation of misfolded proteins in the form of amyloid plaques and neurofibrillary tangles, as well as impaired motor activity with the development of full-blown Alzheimer’s disease dementia are the main phenomena in brain neurodegeneration after ischemia in humans and animals. Due to the pleiotropic effects of myricetin, including anti-amyloid, anti-tau protein, anti-oxidant, anti-inflammatory and anti-dementia properties, myricetin is a promising candidate for the treatment of neurodegeneration following cerebral ischemia (Figure 1). Above effects partially can be explained by increases in sirtuin 1, sirtuin 3, and sirtuin 5 expression in mice treated by myricetin (Figure 2) [153]. Additionally, in this study it was found that myricetin increased mitochondrial content, respiration, and metabolism, and regulated ATP synthesis and cellular ATP generation by an increase in sirtuin 3 activity [153]. The main myricetin molecular mechanisms involved in the protection of neurons by myricetin in post-ischemic brain neurodegeneration are summarized in Figure 2. In addition, it is a safe substance, approved as a pro-health substance, commercially available and inexpensive, which can effectively cross the blood–brain barrier [154]. Holland et al. [154] decided to support the above view to determine whether dietary intake of myricetin in different doses (0.14–1.37 mg/d) by humans is associated with Alzheimer’s disease development and dementia. They presented that the onset of Alzheimer’s disease is inversely associated with dietary intake of myricetin [154]. Higher dietary intake of myricetin may be associated with decreased risk by 38% of the development of Alzheimer’s disease dementia in patients [154]. Recapitulating, the information available in this article about the therapeutic potential of myricetin provides significant evidence for the potential clinical utility of myricetin in the treatment of neurodegenerative disorders with misfolding proteins including post-ischemic brain neurodegeneration [154].

## Figures and Tables

**Figure 1 nutrients-13-00342-f001:**
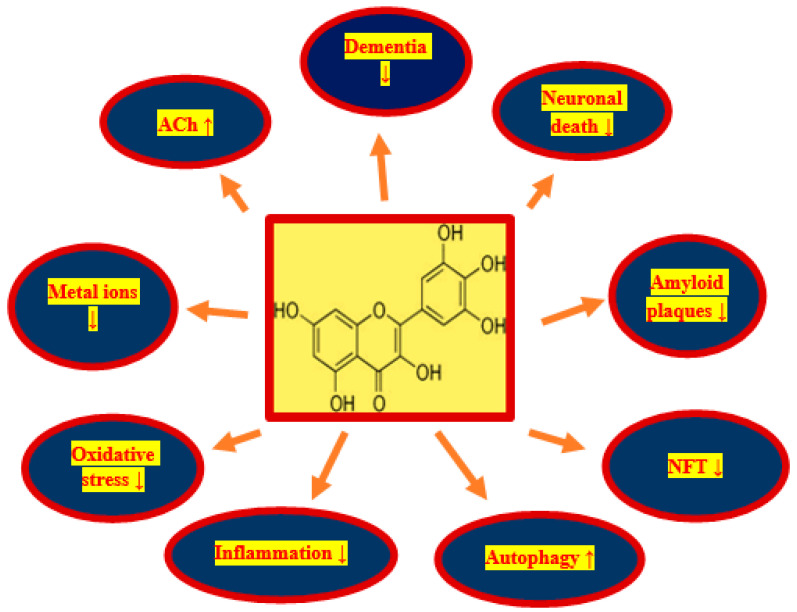
Positive influence of myricetin on phenomena occurring in post-ischemic brain neurodegeneration. In a rectangle structure of myricetin, NFT-neurofibrillary tangles, ACh—acetylcholine, ↓—decrease, ↑—increase.

**Figure 2 nutrients-13-00342-f002:**
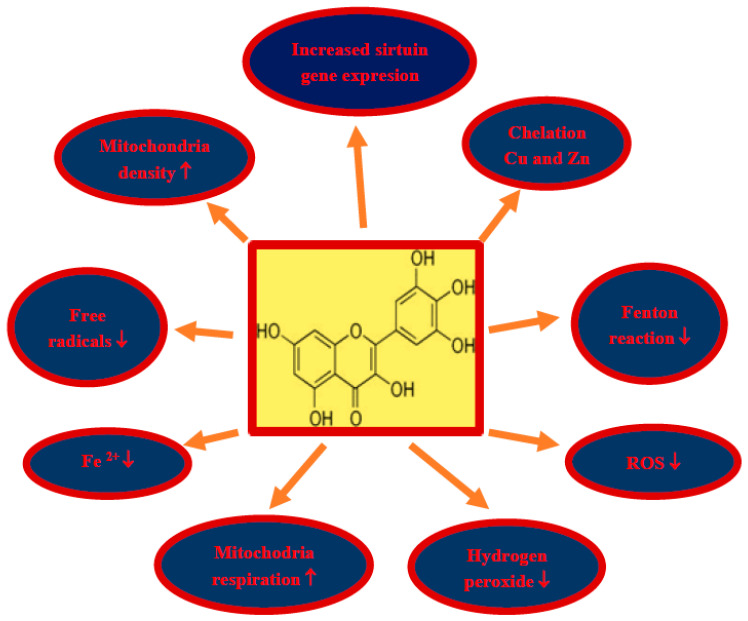
Some molecular mechanisms involved in the protection of neurons by myricetin in post-ischemic brain neurodegeneration. In a rectangle structure of myricetin, ROS—reactive oxygen species, ↓—decrease, ↑—increase.

## Data Availability

Not applicable.

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
