# Peer review of "Myricetin as a Promising Molecule for the Treatment of Post-Ischemic Brain Neurodegeneration"

_nutrients, 2021, doi:10.3390/nu13020342_

Round 1

Reviewer 1 Report

The review article “ Myricetin as a Promising Molecule for the Treatment of 2 Post-ischemic Brain Neurodegeneration” reviews the potential effects of myricetin on post-ischemia neurodegeneration through the following several pathways. (1) reduces the production of amyloid, (2) prevents the development of amyloid oligomers and fibrils, (3) protects the development of neurofibrillary tangles, (3) reduces neuroinflammation, (4) acts as a powerful anti-oxidant, (5) prevents metals from binding to amyloid and tau protein, (6) increases acetylcholine levels. Although the topic is attractive to readers, there are several points that need to be revised.

  1. Many abbreviations must be explained when they first appear in the text.
  2. When citing the reference and providing evidence from the previous studies, the evidence needs to be more specific and detailed. Don’t use words such as “regulate” but use “increase” or “decrease”.
  3. The figure 1 is very simplified. Detail pathways that myricetin works on should be depicted. Such as how myricetin works to enhance autophagy and to inhibit neuroinflammation and oxidative stress should be explained in the figure. I suggest drawing one figure for each pathway.

Author Response

Reviewer 1. The review article “ Myricetin as a Promising Molecule for the Treatment of 2 Post-ischemic Brain Neurodegeneration” reviews the potential effects of myricetin on post-ischemia neurodegeneration through the following several pathways. (1) reduces the production of amyloid, (2) prevents the development of amyloid oligomers and fibrils, (3) protects the development of neurofibrillary tangles, (3) reduces neuroinflammation, (4) acts as a powerful anti-oxidant, (5) prevents metals from binding to amyloid and tau protein, (6) increases acetylcholine levels. Although the topic is attractive to readers, there are several points that need to be revised. 1. Many abbreviations must be explained when they first appear in the text. Done. 2. When citing the reference and providing evidence from the previous studies, the evidence needs to be more specific and detailed. Don’t use words such as “regulate” but use “increase” or “decrease”. Done. 3. The figure 1 is very simplified. Detail pathways that myricetin works on should be depicted. Such as how myricetin works to enhance autophagy and to inhibit neuroinflammation and oxidative stress should be explained in the figure. I suggest drawing one figure for each pathway. The aim of the study was not to present/explain the ultimate molecular mechanisms of myricetin action. The main goal is whether myricetin can be practically used in the prevention or treatment of neuropathological changes and clinical symptoms after ischemic brain neurodegeneration. Unfortunately, if we add extremely complicated molecular mechanisms finally not explained, to the figure 1, which we do not know how these mechanisms are changing in different times during and after ischemia, the figure will not be legible to the ordinary reader, not an expert. The figure is intended to provide each reader with an accessible and simple visual summary of the manuscript. The paper presents numerous possible mechanisms of influence, incl. on changes in amyloid and tau protein, which are important and new pathological elements after brain ischemia. Nevertheless, we added new data in section conclusion and developed an additional second figure based on mechanisms found only partially in the brain after ischemia.

Reviewer 2 Report

This review deals with the multiple effects of the plant flavonoid myricetin on neurons and brain, in the particular case of post-ischemic brain neurodegeneration. It is always difficult to carry out an exhaustive review. The authors chose not to address the molecular mechanisms involved in the protection of neurons by myricetin. These mechanisms of action would help to structure the manuscript, would help to identify prospects for future progress. Here are the main myricetin modes of action poorly treated in the manuscript:

Myricetin prevents rotenone-induced transmembrane electric potential Δψm reduction of mitochondria, essential for respiration yield and ATPase functioning.

Myricetin reducs rotenone-induced ROS production by mitochondria. It is one of the flavonol which target mitochondrial complex I and cytochrome c, inhibit hydrogen peroxide production. These highlight the possibility that myricetin protects cells by anti-oxidative and mitochondrial-protective mechanisms.

Myricetin (1-10 µM) increases cell viability 

Myricetin alleviates the upregulation of hepcidin expression induced by rotenone stress. Hepcidin regulates iron homeostasis by modulating the expression of several iron transporters, avoiding Fenton hydroxyl radical generation.

Myricetin enhances mitochondrial density, induced PGC-1α activity through deacetylation SIRT1 activation.

Myricetin reprogramed gene expression, in particular SIRT3, a critical modulator in mitochondrial metabolism, and delay ageing

To be published in Nutrients, the questions of daily dose efficiency, the myricetin availability and stability in brain should be taken into account. These aspects are totally ignored in the present manuscript. Additional schemes would be useful to highlight the main points.

                For all these raisons, the referee suggests a MINOR REVISION of the manuscript.

Author Response

Reviewer 2. This review deals with the multiple effects of the plant flavonoid myricetin on neurons and brain, in the particular case of post-ischemic brain neurodegeneration. It is always difficult to carry out an exhaustive review. The authors chose not to address the molecular mechanisms involved in the protection of neurons by myricetin. These mechanisms of action would help to structure the manuscript, would help to identify prospects for future progress. Here are the main myricetin modes of action poorly treated in the manuscript: Myricetin prevents rotenone-induced transmembrane electric potential Δψm reduction of mitochondria, essential for respiration yield and ATPase functioning. Myricetin reducs rotenone-induced ROS production by mitochondria. It is one of the flavonol which target mitochondrial complex I and cytochrome c, inhibit hydrogen peroxide production. These highlight the possibility that myricetin protects cells by anti-oxidative and mitochondrial-protective mechanisms. Myricetin (1-10 µM) increases cell viability Myricetin alleviates the upregulation of hepcidin expression induced by rotenone stress. Hepcidin regulates iron homeostasis by modulating the expression of several iron transporters, avoiding Fenton hydroxyl radical generation. Myricetin enhances mitochondrial density, induced PGC-1α activity through deacetylation SIRT1 activation. Myricetin reprogramed gene expression, in particular SIRT3, a critical modulator in mitochondrial metabolism, and delay ageing. The aim of the study was not to explain the ultimate molecular mechanisms of action of myricetin, but the possibility of practical application in the prevention or treatment of neuropathological changes and clinical symptoms in post-ischemic brain neurodegeneration. In the original version of the review, we presented possible mechanisms of influencing, for example, changes in amyloid and tau protein in post-ischemic brain neurodegeneration. The original version of the work also talks about the influence of myricetin on Fenton reaction and iron or ROS as mentioned by the reviewer (see p. 9). Nevertheless, we added the data suggested by the reviewer on the molecular mechanisms of action myricetin in section conclusion, supplementing section references with two references (see 153 and 154). Furthermore, we developed the second figure relying mainly on the molecular mechanisms suggested by the reviewer, for which we thank. To be published in Nutrients, the questions of daily dose efficiency, the myricetin availability and stability in brain should be taken into account. These aspects are totally ignored in the present manuscript. Additional schemes would be useful to highlight the main points. These comments are presented on page 14 of the revised manuscript and in Figure 2. And the literature has been supplemented with two items (see 153 and 154). For all these raisons, the referee suggests a MINOR REVISION of the manuscript.

Round 2

Reviewer 1 Report

The authors have done their best to answer the questions raised.